# Breastfeeding practice, breastfeeding policy and hospitalisations for infectious diseases in early and later childhood: a register-based study in Uppsala County, Sweden

Samuel Videholm [1], Thomas Wallby,[2] Sven-Arne Silfverdal [1]

► Prepublication history and additional online supplemental material for this paper are available online. To view these files, please visit the journal online (http://dx.doi.org/10.1136/bmjopen-2020-046583).

[1] Department of Clinical Sciences, Pediatrics, Umeå University, Umeå, Sweden
[2] Department of Womens and Childrens Health, Uppsala University, Uppsala, Sweden

**Correspondence to**
Dr Samuel Videholm;
samuel.videholm@umu.se

## ABSTRACT

**Objective** To examine the association between breastfeeding practice and hospitalisations for infectious diseases in early and later childhood, in particular, to compare exclusive breast feeding 4–5 months with exclusive breastfeeding 6 months or more. Thereby, provide evidence to inform breastfeeding policy.

**Design** A register-based cohort study.

**Setting** A cohort was created by combining the Swedish Medical Birth Register, the National Inpatient Register, the Cause of Death Register, the Total Population Register, the Longitudinal integration database for health insurance and labour market studies, with the Uppsala Preventive Child Health Care database.

**Patients** 37 825 term and post-term singletons born to women who resided in Uppsala County (Sweden) between 1998 and 2010.

**Main outcome measures** Number of hospitalisations for infectious diseases in early (<2 years) and later childhood (2–4 years).

**Results** The risk of hospitalisations for infectious diseases decreased with duration of exclusive breastfeeding until 4 months of age. In early childhood, breast feeding was associated with a decreased risk of enteric and respiratory infections. In comparison with exclusive breast feeding 6 months or more, the strongest association was found between no breastfeeding and enteric infections (adjusted incidence rate ratios, aIRR 3.32 (95% CI 2.14 to 5.14)). In later childhood, breast feeding was associated with a lower risk of respiratory infections. In comparison with children exclusively breastfed 6 months or more, the highest risk was found in children who were not breastfed (aIRR 2.53 (95% CI 1.51 to 4.24)). The risk of hospitalisations for infectious diseases was comparable in children exclusively breastfed 4–5 months and children exclusively breastfed 6 months or more.

**Conclusions** Our results support breastfeeding guidelines that recommend exclusive breastfeeding for at least 4 months.

## BACKGROUND

Infectious diseases cause large harm and suffering in childhood. Globally, infectious diseases are a leading cause of death in

### Strengths and limitations of this study

► This register-based cohort study included almost all (<10% missing data) term and post-term singletons born in a geographical region over 13 years, which reduces the risk of selection bias.

► Children were followed from birth until 5 years of age or censoring, which enabled us to examine the effect of breast feeding in both early (<2 years) and later childhood (2–4 years).

► Information on pregnancy, birth and sociodemographic characteristics were obtained from high-quality health and administrative registers, which allowed us to adjust analyses for several potential confounders including congenital malformation, large/small for gestational age, maternal age, maternal smoking, parity, maternal education level and maternal country of birth.

► Infections were identified using deidentified hospital discharge data, cases could not be confirmed by information obtained from medical records.

► Breastfeeding practice was not continuously collected, information on breastfeeding practice was only available at 1 week, 2, 4 and 6 months.

children under 5 years of age.[1] In high-income countries, infectious disease remains a major cause of hospitalisations in young children.[2 3] Breast milk contains IgA antibodies and oligosaccharides that prevent microbes to adhere to the mucosa, lactoferrin and lysozyme act directly on the microbes, and growth factors, nucleotides, cytokines stimulate maturation of the infant's immune system.[4 5] Numerous studies have shown that breastfeeding (BF) reduces the risk of infectious diseases in infancy.[4 6 7] A few studies have also reported that BF reduces the risk of infectious diseases after infancy.[8–10]

Although it is well known that BF protects young children in high-income countries against infections, the optimal BF practice is

still under debate. WHO recommends exclusive BF (EBF) until 6 months of age followed by BF along with complementary feeding.[11] The American Academy of Pediatrics recommends BF for at least 12 months with EBF for about 6 months.[12] In contrast, the European Society for Paediatric Gastroenterology, Hepatology, and Nutrition recommend EBF for at least 4 months with EBF or predominant BF for approximately 6 months.[6] The aim of this study was to examine the association between BF practice and hospitalisations for infectious diseases in early (<2 years) and later childhood (2–4 years), in particular, to compare EBF 4–5 months with EBF 6 months or more.

## METHODS
### Patient and public involvement
This study was conducted without patient involvement.

### Setting
In Sweden, all children are offered free primary and hospital care, a free preventive child healthcare programme and a free general immunisation programme; working parents are entitled to a generous parental leave scheme.[13] During the study period, Swedish guidelines for BF duration were consistent with WHO recommendations. Until 2002, EBF was recommended until 4–6 months of age. Thereafter, EBF was recommended for at least 6 months.[14]

### Study population
This is a register-based cohort study of term and post-term (gestational age 37 weeks or more) singletons born to women who resided in Uppsala County between 1998 and 2010. A database was created by combining data from nationwide and local health and administrative registers. The Swedish Medical Birth Register includes information on prenatal, delivery and neonatal care, with coverage of 98%–99% of all births.[15] The National Patient Register contains information on all hospitalisations including International Classification of Disease, 10th Revision (ICD-10) codes for primary diagnosis.[16] The longitudinal integration database for health insurance and labour market studies, includes information on maternal education.[17] The Cause of Death Register provides information on mortality. The Total Population Register contains information about migration and maternal country of birth. The Child Healthcare Quality database in Uppsala contains data on BF, which was collected by child health nurses during free routine health checkups.[13] These registers were linked using the unique personal identification number assigned to every Swedish resident, by the Centre for Epidemiology at the Swedish National Board of Health and Welfare. All data were anonymised and deidentified prior to analysis.

### Main exposures
BF was categorised according to intensity and duration. Information on current BF status (EBF, partial BF or no

BF) was collected at 1 week, 2, 4 and 6 months. BF was categorised as no BF if 'no BF' was reported at 1 week. Thereafter, EBF was considered to be present until "partial BF" or "no BF" was reported; BF was considered to be present until 'no BF' was reported, for example, BF was categorised as 'EBF 4–5 months with BF ≥6 months' if 'EBF' was reported at 1 week, 2 months, 4 months and 'partial BF' was reported at 6 months. Information on BF intensity and duration was combined into the following categories: no BF, EBF <4 months with BF <6 months (EBF <4 months (MO) with BF <6 MO), EBF <4 months with BF ≥6 months (EBF <4 MO with BF ≥6 MO), EBF 4–5 months with BF <6 months (EBF 4–5 MO with BF <6 MO), EBF 4–5 months with BF ≥6 months (EBF 4–5 MO with BF ≥6 MO) and EBF ≥6 months (EBF ≥6 MO). Before 2004, the Swedish definition of EBF allowed "small tastes" that did not replace a breast milk feed. In 2004 the WHO definition of EBF was adopted in Sweden, and tastes were no longer accepted to fulfil this criterion.[18 19]

### Covariates
Analyses were adjusted for several child and maternal characteristics. Information on small for gestational age (yes or no), large for gestational age (yes or no), congenital malformation (ICD-10 codes Q00-Q99) and sex (male or female), maternal age (≤19, 20–24, 25–29, 30–34 and ≥35), parity (1, 2, 3 and ≥4), maternal smoking at the first antenatal care visit (yes or no) and year of birth were retrieved from the Medical Birth Register. Maternal education level at year of birth (secondary school or less (≤9 years), upper secondary school (10–12 years), short postsecondary education (13–14 years) and long postsecondary education (≥15 years)) was obtained from the Longitudinal integration database for health insurance and labour market studies. Information on maternal country of birth (Sweden, Other Nordic, Other Europe and North America, Asia, Africa and Other) was retrieved from the Total Population Register.

### Outcomes
The main outcome was overall number of hospitalisations with a principal diagnosis of infectious disease recorded in the National Patient Register. Secondary outcomes were number of hospitalisations for respiratory tract and enteric infections. Hospitalisations were recorded using ICD-10 codes. We used a previously developed coding scheme to identify infectious disease codes and group them into infectious disease categories (online supplemental appendix A).[3]

### Statistical analysis
Crude and adjusted associations between BF practice and number of hospitalisations were estimated using negative binomial regression models. Log follow-up time (days) was used as an offset. Results were presented as adjusted incidence rate ratios (aIRRs) with 95% CIs. Correlations between siblings were accounted for using generalised estimating equations with robust standard errors.[20] All

analyses were adjusted for time trends (year of birth). The adjusted analyses were also controlled for small for gestational age, large for gestational age, congenital malformation, sex, maternal age, parity, maternal smoking, maternal education level and maternal country of birth. Crude and adjusted models were fitted for each outcome (overall, respiratory and enteric infections) and follow-up period (early childhood (<2 years) and later childhood (2–4 years)). In the later follow-up period, an additional adjusted model controlled for previous admissions (a binary variable indicating hospitalisation in the early follow-up period) was fitted for each outcome. All models were restricted to observations with complete data on BF and covariates. In sensitivity analyses, the respiratory category was divided into upper and lower respiratory infections, analyses were adjusted for maternal body mass index (BMI) during early pregnancy and analyses were stratified by year of birth (1998–2003 and 2004–2010). Maternal BMI was excluded from the original analyses due to a large proportion of missing data (12%). All statistical analyses were performed using Stata V.14 (Stata, 2015. Stata Statistical Software: Release 14).

## RESULTS

The Swedish Medical Birth Register included 41 825 term and post-term singletons born to women who resided in Uppsala County between 1998 and 2010. We excluded children with missing data on BF (n=2127) or covariates (n=1873), leaving 37 825 children (90% of the original cohort). In the first follow-up period (early childhood), all children were followed from birth until 2 years of age or censoring due to death (n=10) or emigration (n=194). In the later follow-up period (later childhood), the remaining 37 621 children were followed from 2 until 5 years of age or censoring due to death (n=14) or emigration (n=573). The study included 187 763 person-years (PY) of follow-up time, 75 513 PY in early childhood and 112 250 PY in later childhood. In total 4728 hospital admissions for infectious diseases were recorded during the study period. Readmissions on the same day recorded with the same infectious disease group were excluded (n=23) leaving 4705 admissions. In early childhood, the study included 1485 admissions for respiratory infections (799 admissions for upper respiratory infections and 686 admissions for lower respiratory infections), 762 admissions for enteric infections and 1080 admissions for other infections. In later childhood, the study included 630 admissions for respiratory infections (381 admissions for upper respiratory infections and 249 admissions for lower respiratory infections), 300 admissions for enteric infections and 448 admissions for other infections.

Table 1 presents maternal and child characteristics. Most mothers initiated BF, only 1.6% (n=613) of mothers reported no BF at 1 week. A majority of women exclusively breastfed for at least 4 months, EBF 4–5 months with BF <6 months was reported by 3.9%, EBF 4–5 months with BF ≥6 months was reported by 40% and EBF ≥6 months

was reported by 25%. Women over 25 years of age and women with a postsecondary education, were more likely to report EBF for 4 months or more. In contrast, women who reported smoking during pregnancy were more likely to report no BF at 1 week.

Figure 1 shows the association between BF categories and overall hospitalisations for infectious disease in early and later childhood. The risk of hospitalisations for infections decreased with duration of EBF until 4 months of age. In comparison with children exclusively breastfed 6 months or more, the highest risk of hospitalisations for infectious diseases in early childhood was found in children who were not breastfed (aIRR 1.89 (95% CI 1.45 to 2.47)) and in children exclusive breastfed <4 months with BF <6 months (aIRR 1.41 (95% CI 1.25 to 1.59)). Similarly, the risk of hospitalisations for infectious diseases in later childhood was highest in children who were not breastfed (aIRR 1.52 (95% CI 1.00 to 2.33), controlled for previous admissions) and in children breastfed <6 months and EBF <4 months (aIRR 1.43 (95% CI 1.16 to 1.75), controlled for previous admissions). The risk of hospitalisations for infectious diseases was comparable in children exclusively breastfed 4 to 5 months and children exclusively breastfed 6 months or more. Online supplemental appendix B includes full regression results including sensitivity analyses.

Figure 2 shows associations between BF categories and hospitalisations for respiratory tract and enteric infections in early and later childhood. In early childhood, the risk of both respiratory and enteric infections decreased with duration of EBF until 4 months of age. In comparison with EBF 6 months or more, the strongest association was found between no BF and enteric infections (aIRR 3.32 (95% CI 2.14 to 5.14)). In later childhood, the risk of respiratory infections decreased with duration of EBF until 4 months of age. In comparison with children exclusively breastfed 6 months or more, the highest risk of respiratory infections was observed in children who were not breastfed (aIRR 2.53 (95% CI 1.51 to 4.24), controlled for previous admissions). Online supplemental appendix C includes full regression results including sensitivity analyses.

Sensitivity analyses showed no substantial changes in point estimates after adjusting for pregnancy BMI. Moreover, sensitivity analyses showed similar associations between BF categories and overall hospitalisations for infectious disease in the first (1998–2003) and second (2004–2010) birth cohort. In sex-stratified analyses, associations between BF categories and overall hospitalisations for infectious disease were similar in boys and girls (data not shown).

## DISCUSSION

We found that the risk of overall hospitalisations for infectious diseases in early childhood (<2 years) and the risk of hospitalisations for respiratory infections in later childhood (2–4 years) decreased with duration of EBF until

**Table 1** Child and maternal characteristics and their association with BF categories

| | Not BF | EBF <4 MO with BF <6 MO | EBF <4 MO with BF ≥6 MO | EBF 4–5 MO with BF <6 MO | EBF 4–5 MO with BF ≥6 MO | EBF ≥6 MO | All included | |
|---|---|---|---|---|---|---|---|---|
| | % (n) | % (n) | % (n) | % (n) | % (n) | n | n | Excluded |
| All | 1.6 (613) | 19 (7335) | 11 (3998) | 3.9 (1488) | 40 (15007) | 25 (9384) | 37825 | 4000 |
| SGA | | | | | | | | |
| Yes | 1.5 (8) | 27 (144) | 16 (86) | 4.1 (22) | 32 (170) | 20 (106) | 536 | 85 |
| No | 1.6 (605) | 19 (7191) | 11 (3912) | 3.9 (1466) | 40 (14837) | 25 (9278) | 37289 | 3816 |
| LGA | | | | | | | | |
| Yes | 2.5 (43) | 22 (385) | 11 (189) | 3.5 (62) | 35 (614) | 26 (457) | 1750 | 172 |
| No | 1.6 (570) | 19 (6950) | 11 (3809) | 4.0 (1426) | 40 (14393) | 25 (8927) | 36075 | 3729 |
| Congenital malformation* | | | | | | | | |
| Yes | 4.0 (32) | 25 (196) | 9.7 (77) | 3.6 (29) | 35 (277) | 23 (186) | 797 | 3891 |
| No | 1.6 (581) | 19 (7139) | 11 (3921) | 3.9 (1459) | 40 (14730) | 25 (9198) | 37028 | 109 |
| Sex | | | | | | | | |
| Male | 1.6 (319) | 20 (3926) | 11 (2077) | 4.1 (795) | 40 (7751) | 24 (4731) | 19599 | 2088 |
| Female | 1.6 (294) | 19 (3409) | 11 (1921) | 3.8 (693) | 40 (7256) | 26 (4653) | 18226 | 1912 |
| Maternal age | | | | | | | | |
| ≤19 | 2.5 (10) | 47 (187) | 7.5 (30) | 8.2 (33) | 24 (98) | 11 (43) | 401 | 146 |
| 20-–24 | 1.6 (67) | 33 (1402) | 9.0 (378) | 6.7 x (282) | 32 (1362) | 17 (723) | 4214 | 607 |
| 25–29 | 1.7 (198) | 21 (2366) | 9.8 (1136) | 4.7 (545) | 40 (4582) | 24 (2727) | 11554 | 1167 |
| 30–34 | 1.6 (222) | 15 (2108) | 11 (1482) | 3.4 (470) | 43 (5876) | 27 (3682) | 13840 | 1305 |
| ≥35 | 1.5 (116) | 16 (1272) | 12 (972) | 2.0 (158) | 40 (3089) | 28 (2209) | 7816 | 775 |
| Parity | | | | | | | | |
| 1 | 1.1 (185) | 22 (3510) | 12 (1878) | 3.9 (642) | 39 (6384) | 23 (3724) | 16323 | 1821 |
| 2 | 1.9 (268) | 18 (2486) | 9.3 (1301) | 4.3 (603) | 41 (5738) | 26 (3584) | 13980 | 1336 |
| 3 | 2.1 (110) | 17 (902) | 10 (541) | 3.1 (165) | 41 (2177) | 27 (1440) | 5335 | 535 |
| ≥4 | 2.3 (50) | 20 (437) | 13 (278) | 3.6 (78) | 32 (708) | 29 (636) | 2187 | 308 |
| Maternal smoking | | | | | | | | |
| No smoking | 1.5 (521) | 18 (6136) | 11 (3744) | 3.8 (1337) | 41 (14362) | 26 (9061) | 35161 | 2366 |
| Smoking | 3.5 (92) | 45 (1199) | 9.5 (254) | 5.7 (151) | 24 (645) | 12 (323) | 2664 | 255 |
| Maternal education† | | | | | | | | |
| ≤9 | 2.9 (102) | 36 (1273) | 11 (406) | 5.4 (193) | 29 (1014) | 16 (561) | 3549 | 419 |
| 10–12 | 2.2 (338) | 25 (3942) | 10 (1590) | 5.4 (838) | 35 (5516) | 22 (3378) | 15602 | 1217 |
| 13–14 | 1.1 (56) | 15 (725) | 10 (500) | 3.2 (161) | 42 (2071) | 30 (1472) | 4985 | 491 |
| ≥14 | 0.9 (117) | 10 (1395) | 11 (1502) | 2.2 (296) | 47 (6406) | 29 (3973) | 13689 | 1217 |
| Maternal country of birth | | | | | | | | |
| Sweden | 1.6 (526) | 19 (6231) | 9.9 (3168) | 4.1 (1313) | 40 (12853) | 25 (7977) | 32068 | 2693 |
| Other Nordic | 0.8 (6) | 17 (134) | 8.8 (70) | 2.4 (19) | 41 (326) | 30 (238) | 793 | 128 |
| Other Europe and North America | 1.8 (27) | 20 (303) | 12 (185) | 3.2 (48) | 37 (564) | 25 (382) | 1509 | 339 |
| Asia | 1.3 (33) | 22 (538) | 16 (399) | 3.0 (75) | 36 (895) | 21 (528) | 2468 | 540 |
| Africa | 1.9 (12) | 11 (67) | 20 (126) | 3.1 (19) | 38 (233) | 26 (163) | 620 | 224 |

Continued

**Table 1** Continued

| | Not BF | EBF <4 MO with BF <6 MO | EBF <4 MO with BF ≥6 MO | EBF 4–5 MO with BF <6 MO | EBF 4–5 MO with BF ≥6 MO | EBF ≥6 MO | All included | |
|---|---|---|---|---|---|---|---|---|
| | % (n) | % (n) | % (n) | % (n) | % (n) | n | n | Excluded |
| Other | 2.5 (9) | 17 (62) | 14 (50) | 3.8 (14) | 37 (136) | 26 (96) | 367 | 73 |

Children with missing data on BF (n=2127) or covariates (n=1873) were excluded leaving 37 825 children.
*ICD-10 codes: Q00–Q99.
†Maternal education level at year of birth, categorised as 9 years or less, 10–12 years, 13–14 years or 15 years or more.
BF, breast feeding; EBF, exclusive BF; LGA, large for gestation age; MO, months; SGA, small for gestation age.

4 months of age. In comparison with EBF 6 months or more, EBF 4–5 months was not associated with a higher risk of enteric or respiratory infections.

There is strong evidence that BF protects infants from infectious diseases.[4 6 7] However, the optimal BF practice is under discussion. Our results are consistent with a Spanish study reporting that full BF until at least 4 months would prevent 56.4% (95% CI 30.9% to 69.4%) of non-perinatal infection in the first year of life.[21] In comparison with no BF, a Dutch study found that EBF until 4 months followed by partially BF decreased the risk of physician-confirmed upper respiratory (adjusted OR (aOR) 0.65 (95% CI 0.51 to 0.83)) lower respiratory (aOR 0.50 (95% CI: 0.32 to 0.79)) and gastrointestinal infections (aOR 0.41 (95% CI 0.26 to 0.64)) during the first 6 months. However, the authors concluded that EBF until 6 months may reduce the risk of infectious diseases even more.[22] In contrast to our findings, a US study reported that full BF for 6 months was, in comparison with full BF for 4 months, associated with a decreased risk of respiratory tract infection during the first 2 years of life.[23] Moreover, observational analysis of the 'Promotion

of Breastfeeding Intervention Trial' found that EBF for 6 months was, compared with EBF for 3 months and partial BF thereafter, associated with a decreased risk of gastrointestinal episodes; no significant association was found for gastrointestinal hospitalisations.[7] The Dutch and US studies included less severe infections, whereas our study only included infections requiring hospital care.

There is increasing evidence that BF also reduces the risk of respiratory infections after infancy. The mechanism is still unknown, however, it has been suggested that breastmilk influence the development of the immune system.[8 24] Our findings are similar to those reported in a recent Japanese study. In comparison with no BF, EBF at 6–7 months of age was associated with a reduced risk of hospitalisations for respiratory tract infections (aOR 0.76 (95% CI: 0.58 to 0.99)) between 30 and 42 months of age, whereas no association was found between BF and hospitalisations for diarrhoea.[9] In a Dutch cohort study, BF for at least 6 months was in comparison with no BF associated with a decreased risk lower respiratory tract infections (aOR 0.71 (95% CI: 0.51 to 0.98)) between infancy and 4 years of age.[10] Moreover, a recent US study of children

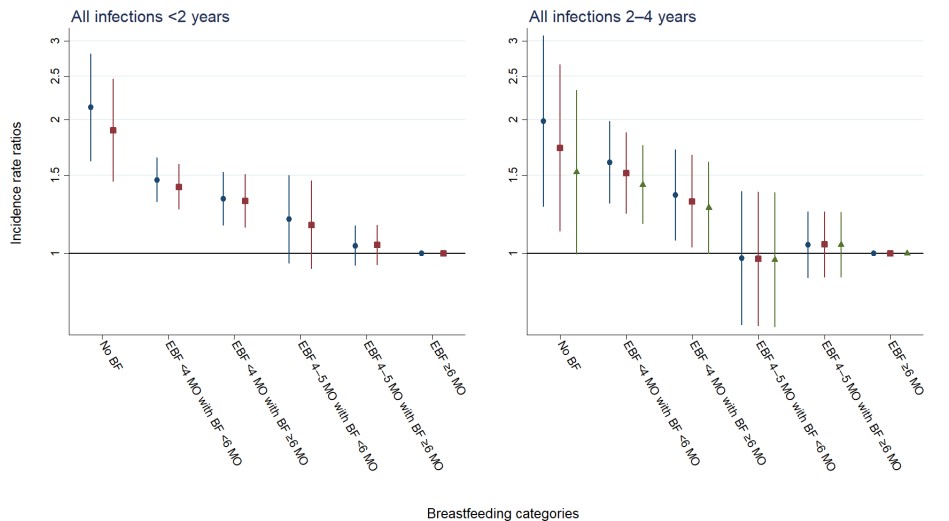

**Figure 1** Crude analyses (blue circles), adjusted analyses (red squares) and adjusted analyses controlled for previous admissions (green triangles) of the associations between BF categories and overall hospitalisations for infectious disease, in early childhood (<2 years) and later childhood (2–4 years). Incidence rate ratios are presented on a logarithmic scale. Vertical lines represent 95% CIs around the point estimates. BF, breast feeding, EBF, exclusive BF; MO, months.

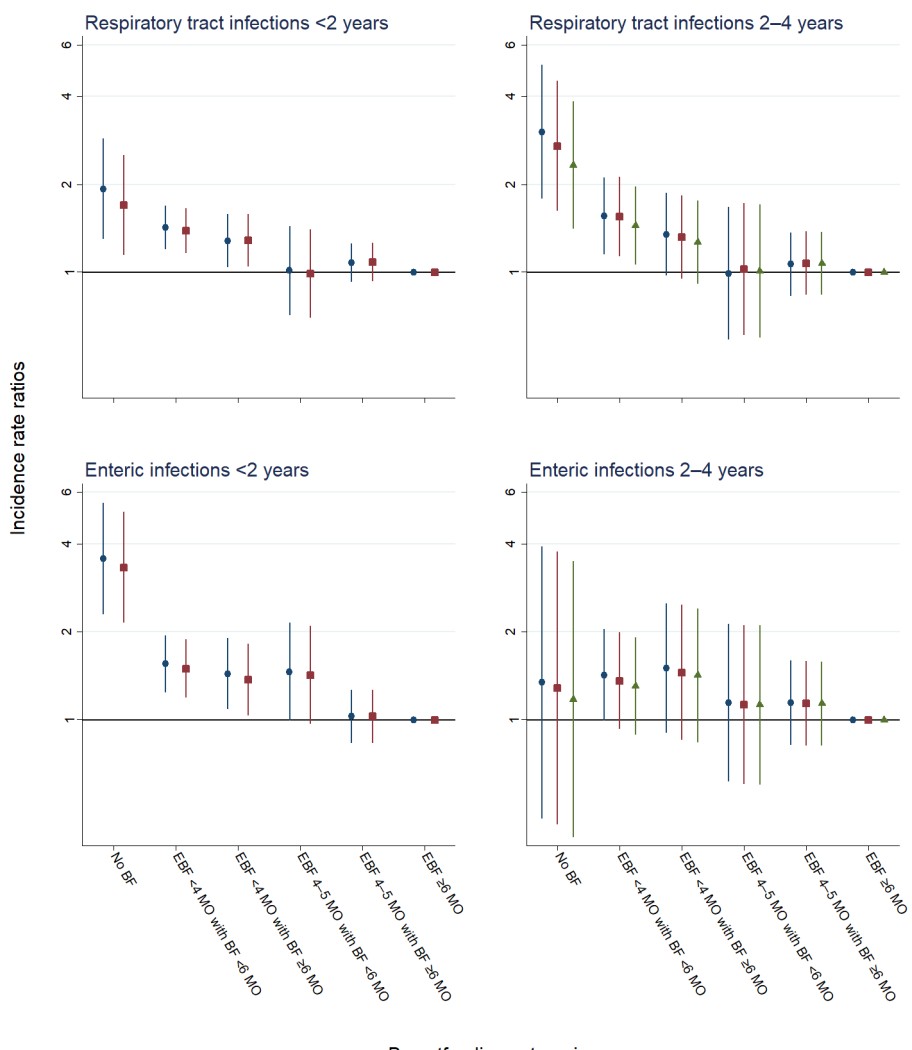

**Figure 2** Crude analyses (blue circles), adjusted analyses (red squares) and adjusted analyses controlled for previous admissions (green triangles) of the associations between BF categories, and hospitalisations for respiratory tract and enteric infections, in early childhood (<2 years) and later childhood (2–4 years). Incidence rate ratios are presented on a logarithmic scale. Vertical lines represent 95% CIs around the point estimates. BF, breast feeding; EBF, exclusive BF; MO, months.

aged 6 years, found that any BF for 9 months or more was in comparison with any BF for >0 to <3 months associated with a decreased risk of ear (aOR 0.69 (95% CI 0.48 to 0.98)), throat (aOR 0.68 (95% CI 0.47 to 0.98)), and sinus (aOR 0.47 (95% CI 0.30 to 0.72)) infections, but not with cold/upper respiratory tract infections and pneumonia.[8] Overall, these findings are consistent with our results, that BF reduces the risk of respiratory infections after infancy.

Our study has several strengths. First, the use of high-quality health and administrative registers allowed us to adjust analyses for potential confounders including congenital malformation, large/small for gestational age, maternal age, maternal smoking, maternal education level and maternal country of birth. Second, the long follow-up period (until 5 years of age or censoring) enabled us to examine the effect of BF in both early and

later childhood. Additionally, this allowed analyses in later childhood to be adjusted for hospitalisations in early childhood. Finally, our study included almost all children (<10% missing data) in one county, thereby reducing the risk of selection bias. However, our study has several weaknesses. In early childhood, infections episodes may influence BF practice leading to reverse causation. In later childhood, confidence intervals were wide due to the small number of events. Therefore, the associations between BF and infections in later childhood need to be interpreted with caution. Information on current BF status was not continuously collected. Compared with studies with complete information on BF practice since birth, for example, collected through daily recordings, our study is likely to overestimate the proportion of exclusively breastfed children.[25] Moreover, Sweden has family-friendly policies that enable BF including a generous

parental leave scheme.[13] Consequently, our findings may not be generalised to other populations with less family-friendly policies and lower BF rates. Finally, due to the observational nature of this study, we cannot rule out the risk of unmeasured or residual confounding.

## CONCLUSIONS

Our study found that the risk of overall hospitalisations for infectious diseases in early childhood and the risk of hospitalisations for respiratory infections in later childhood decreased with duration of EBF until 4 months of age. Additionally, the risk of hospitalisations for infectious diseases was comparable in children exclusively breastfed 4–5 months and children exclusively breastfed 6 months or more. Thereby, it supports the current European Society for Paediatric Gastroenterology, Hepatology and Nutrition guidelines, recommending EBF for at least 4 months with exclusive or predominant BF for approximately 6 months. Moreover, it adds to the growing body of evidence suggesting a protective effect of BF on respiratory infections after infancy.

**Acknowledgements** The authors would like to thank Anna Lindam, PhD at the Unit of Research, Development, and Education, Östersund Hospital, Sweden, for assistance with statistical analysis and Håkan Sjöberg, MEc at Statistics Sweden, Örebro, Sweden, for help in compiling the dataset.

**Contributors** SV conceptualised and designed the study together with S-AS, carried out analyses and drafted the initial manuscript. TW supervised the collection of breastfeeding data, contributed to the study design, reviewed and revised the manuscript. S-AS conceptualised and designed the study together with SV, coordinated and supervised the database, reviewed and revised the manuscript.

**Funding** This study was supported by the Unit of Research, Development, and Education, Östersund Hospital, Östersund, Sweden (JLL-930202 (to SV)); ALF Umeå University, Umeå, Sweden (RV-933162 (to S-AS)).

**Competing interests** None declared.

**Patient consent for publication** Not required.

**Ethics approval** The project was approved by the Regional Ethics Board in Umeå (reference number 2012-265-31M and 2017-399-32M).

**Provenance and peer review** Not commissioned; externally peer reviewed.

**Data availability statement** Data may be obtained from a third party and are not publicly available. We used deidentified register data obtained from third parties. It includes sensitive information and some access restrictions may apply. Interested researchers need to obtain data directly from the National Board of Health and Welfare in Sweden (socialstyrelsen@socialstyrelsen.se), from Statistics Sweden ( scb@scb.se) and from the Child Health Service Unit in Uppsala (thomas.wallby@ kbh.uu.se). Children included in the study were identified in the Medical Birth Register, data on hospitalisations were obtained from the Swedish National Patient Register and data on deaths were obtained from the Cause of Death Register. All of these registers are maintained by the National Board of Health and Welfare in Sweden. Data on maternal education were obtained from the Longitudinal Integration Database for Health Insurance and Labour market Studies and data on migration were obtained from The Total Population Register, both registers are maintained by Statistics Sweden. Data on breast feeding was obtained from The Child Health Care Quality database, which is maintained by the Child Health Service Unit in Uppsala.

**ORCID iDs**
Samuel Videholm http://orcid.org/0000-0002-1468-5771
Sven-Arne Silfverdal http://orcid.org/0000-0002-3606-3797

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
