## [Reviewer comments · BMJ Open]

This paper was submitted to a another journal from Archives of Disease in Childhood but declined for publication following peer review. The authors addressed the reviewers' comments and submitted the revised paper to BMJ Open. The paper was subsequently accepted for publication at BMJ Open.

(This paper received three reviews from its previous journal and three reviewers agreed to published their review.)

ARTICLE DETAILS

TITLE (PROVISIONAL)	Breastfeeding practice, breastfeeding policy and hospitalisations for infectious diseases in early and later childhood: a register-based study in Uppsala County, Sweden
AUTHORS	Videholm, Samuel; Wallby, Thomas; Silfverdal, Sven-Arne

VERSION 1 – REVIEW

REVIEWER	Bjerregaard, Lise Center for Clinical Research and Disease Prevention, Bispebjerg and Frederiksberg Hospital
REVIEW RETURNED	20-Jul-2020

GENERAL COMMENTS	The aim of the study is to examine the associations between breastfeeding practices and hospitalisations for infectious diseases in the first five years of life. The conclusions are that exclusive breastfeeding at least until four months was associated with a lower risk of overall hospitalisations for infectious diseases in early childhood. Additionally, breastfeeding was associated with a decreased risk of hospitalisations for respiratory infections in later childhood. This is a very relevant and interesting study and it is based on a unique large dataset with rich information. The statistical analyses are appropriate, but some issues are not entirely clear. The manuscript is in general well-written. I have several comments, none really major, and some suggestions for the authors to consider: 1. The description of the BF group is not easily understood. I suggest that the author delete "or without" in the description of the BF groups: "breastfed < 6 months with or without exclusive breastfeeding < 4 months" and "and breastfed ≥ 6 months with or without exclusive breastfeeding < 4 months". Then the descriptions will correspond to the abbreviations used by the authors: "BF < 6 MO and EBF <4 MO" and "BF ≥ 6 MO and EBF <4 MO".2. Abstract result/conclusion: Since the aim is to contribute with knowledge for the recommendation of 4 or 6 months of EBF, it would strengthen the relevance of the study, if it would be possible to include the finding that EBF 4-5 mo is not associated with a reduced risk.
--

3. P. 5 main exposure: How was missing info on BF handled? Was the last information brought forward or were the duration truncated at the last measurement available? How many measures did the kids have on average? How many were followed until age 2, 4 and 6 mo? And were kids included if they only had a measurement at week 1? The BF pattern at week 1 may not be so informative, taking into account that almost all women initiate BF. Did children who were excluded due to missing information on BF differ from those included? Maybe these things could be described in an appendix.

4. P. 6: Covariates: How were covariates chosen? I wonder if all of these are relevant confounders? Are they associated with BF and the outcomes? Why did the authors group maternal age and parity instead of using these as continuous variables? Similarly for year of birth.

5. Could there be an interaction with Sex? Since boys and girls have different risks of infections in infancy? Did the authors investigate sex-specific analyses?

6. What is the number of cases for enteric and upper/lower respiratory infections? It would be preferable to have them in the manuscript, but they could also be added to the supplementary tables.

7. P. 8 line 52-56 + P 9 line 29-31 + p. 11 line 40 + p. 12 line 25: I slightly disagree with the authors in the interpretation that, in later childhood, the risk for overall hospitalization remains higher only for children BF < 6 mo with EBF <4 mo. For the two other groups who had reduced risks in early childhood, the lower 95% CI for IRR is 1.00 – and the effect size estimate (the value most compatible with the data) is still clearly positive and not so much different from in early childhood.

8. P 8: I think the authors need to mention that children EBF for 4-5 mo did not have different risks than those EBF for 6 mo. This is an important finding in terms of providing evidence for BF recommendations.

9. P. 8 line 56-58: It would be preferable if the authors describe what the appendix shows (here or in the appendix). Did adjustment for BMI change the results?

10. P 9 line 5-12: + p. 9 line 36 + p 11 line 43 + p. 12 line 30: I do not follow why the authors only report/mention the association between BF and respiratory tract infection in later childhood. Although the associations in early childhood are weaker, they are all significant (if EBF < 4 mo) and follow the same pattern as in later childhood.

11. P. 10: line 3-7: I do not follow how the findings from the US study contracts with the results by the authors. This study also showed an association between BF and respiratory tract infection in early childhood.

12. Nice figures. The authors may consider showing the y-axis on logarithmic scale. Then it will be evident than an IRR of 0.5 is of similar strength as an IRR of 2.

Minor

- P. 6: what is "LISA"?
- P. 7: Please describe the time unit for the statistical analyses.
- P. 8 and elsewhere: I think the authors mean 'censoring' rather than 'censuring'.
- P. 8 line 41+: it is not clear if the authors present crude or adjusted results.
- P. 9 line 34 + p. 11 line 40: Do the authors mean 'exclusive bf for at least 4 months was associated with a reduced risk of hosp. for respiratory infections..'? (i.e. add exclusive)
- p. 11 line 25: Do the authors mean "lack of any bf and enteric

	infections in later childhood"? there as an association between BF and respiratory tract infection in later childhood. - p. 37, Table G: first line: Please check the estimate of IRR=0.01 (0.01-0.01). This seems incorrect. How many cases are in the group?
--	---

REVIEWER	Cole, Tim UCL Institute of Child Health
REVIEW RETURNED	20-Jul-2020

GENERAL COMMENTS	The authors show that hospital admissions for infection in young people are inversely associated with their breastfeeding exposure. I have some comments on the study analysis, presentation and interpretation.  1. Breastfeeding exposure is defined in terms of months of breastfeeding and months of exclusive breastfeeding, in quite a complicated way as described at the top of page 6. It needs to be stated there that the six groups ranked in the order they are defined represent increasing exposure to breastmilk, and that this can be exploited in the analysis. 2. If the groups are numbered from 0 to 5 (or equivalently 1 to 6) then the results in Figures 1 and 2 can be tested for their association with breastmilk exposure by adding the interaction term to the model, which is evident in most of the plots. It is important to view the results "in the round" rather than picking out particular groups that are or are not significant - the trend is what matters. These results would be best presented in a table, perhaps comparing the fit of the models with a linear trend to those with the data grouped. 3. Some smaller points. The Abstract Results involve lots of aIRRs which are hard to read, and presenting them to one not two decimal places would help the reader without losing important information. Perhaps also make the point there that they represent a dose-response relationship. 4. The Introduction includes the unreferenced statement that "Breastfeeding reduces the risk of infectious diseases", which looks odd as the study aims to test the statement. I recommend omitting the sentence, and it then reads well. 5. The acronym LISA is not defined. 6. The country groupings can be simplified to: Sweden, Other Nordic countries, Other Europe or North America, Asia, Africa and other (see page 6 and Table 1). 7. The negative binomial models have one outcome measure and so are univariate; in addition they have multiple covariates and so are multivariable. 8. In the Results several numbers have commas rather than periods as delimiter. 9. The word 'censuring' should be 'censoring' in several places. Also 'emigration' is clearer than 'international migrating'. 10. Table 1 would be easier to read with the following changes: add a Total row at the top, giving numbers and percentages in each of
--

	the six groups, and the percentages can then be compared with those in later rows to see if the group is over- or under-represented. Also the numbers and percentages would be better formatted as % (n) with the percentages given to two significant digits, i.e. whole numbers for 10% or more. 11. In Figures 1 and 2 IRR should be on a log scale, i.e. 2 and 0.5 should be equidistant from 1. This will have the effect of making the plots closer to linear and hence emphasising the linear trend. 12. Personally I am not convinced that the multiple supplementary tables add much of value. However if they are retained they should include information about the linear trend, or even replace the grouped results with the trend. Tim Cole
--	---

REVIEWER	Wong, Peter Hospital for Sick Children
REVIEW RETURNED	20-Jul-2020

GENERAL COMMENTS	Summary: This observational cross-sectional study to examine the associations between breastfeeding practices and hospitalisations for infectious disease in the first five years of life. This large cohort from Sweden between 1998 and 2010 included over 37,000 full-term singleton infants. Authors concluded that exclusive breastfeeding for at least 4 months was associated with a lower risk of overall hospitalisations for infectious diseases in early childhood and Breastfeeding was associated with decreased risk of hospitalisation for respiratory illness in later childhood. Major comments:  1. The study cohort is large from Sweden. The findings are not unique and well described in other association studies; however, this study does serve to strengthen current knowledge. 2. Strongest determinants of both breastfeeding practice and infection risk is low socioeconomic status and poverty. Covariates of study including maternal education, maternal country of birth and maternal smoking may not adequately adjust for low socioeconomic status and/or low neighbourhood equity. Recommend adding to limitations section. 3. As noted in Setting, breastfeeding guidelines in Sweden changed during the study period (1998-2002 EBF 4-6 months and 2002-2010 EBF >6months). These guidelines may have affected the duration of EBF of participants, findings and interpretation of results. Recommend adjusting for periods and/or outlining in limitation section. Minor comments:  1. Study may not be generalisable to other populations; especially since Sweden has relatively high breastfeeding rates, free primary and health care, and generous parental leave scheme, compared to other industrialized countries. Recommend outlining in limitation section. 2. Other limitations may include unmeasured confounders, such as seasonal and cyclical variation of infectious disease (Inparticular RSV which impacts young children) and Indigenous peoples of Sweden (may have lower BF rates and increased rates of RSV
---

VERSION 1 – AUTHOR RESPONSE

Comments Archives of Disease in Childhood:

Reviewer(s)' Comments to Author:

Reviewer: 1

Comments to the Author

The aim of the study is to examine the associations between breastfeeding practices and hospitalisations for infectious diseases in the first five years of life.

The conclusions are that exclusive breastfeeding at least until four months was associated with a lower risk of overall hospitalisations for infectious diseases in early childhood. Additionally, breastfeeding was associated with a decreased risk of hospitalisations for respiratory infections in later childhood.

This is a very relevant and interesting study and it is based on a unique large dataset with rich information. The statistical analyses are appropriate, but some issues are not entirely clear. The manuscript is in general well-written. I have several comments, none really major, and some suggestions for the authors to consider:

1. The description of the BF group is not easily understood. I suggest that the author delete “or without” in the description of the BF groups: “breastfed < 6 months with or without exclusive breastfeeding < 4 months” and “and breastfed \geq 6 months with or without exclusive breastfeeding < 4 months”. Then the descriptions will correspond to the abbreviations used by the authors: “BF < 6 MO and EBF <4 MO” and “BF \geq 6 MO and EBF <4 MO”.

Reply: The text has been revised as suggested:

“No breastfeeding (No BF), breastfeeding <6 months and exclusive breastfeeding <4 months (BF <6 MO and EBF <4 MO), breastfeeding <6 months with exclusive breastfeeding 4–5 months (BF <6 MO with EBF 4–5 MO), breastfeeding \geq 6 months and exclusive breastfeeding <4 months (BF \geq 6 MO and EBF <4 MO), breastfeeding \geq 6 months with exclusive breastfeeding 4–5 months (BF \geq 6 MO with EBF 4–5 MO) and exclusive breastfeeding \geq 6 months (EBF \geq 6 MO).”

2. Abstract result/conclusion: Since the aim is to contribute with knowledge for the recommendation of 4 or 6 months of EBF, it would strengthen the relevance of the study, if it would be possible to include the finding that EBF 4-5 mo is not associated with a reduced risk.

Reply: The text has been revised as suggested:

Abstract: “In comparison with breastfeeding 4 to 5 months, exclusively breastfeeding 6 months or more was not associated with a lower risk of enteric or respiratory infections.”

Results: “The risk of hospitalisations for infectious diseases was comparable in children exclusively breastfed 4 to 5 months and children exclusively breastfed 6 months or more.”

Conclusion “Additionally, the risk of hospitalisations for infectious diseases was comparable in children exclusively breastfed 4 to 5 months and children exclusively breastfed 6 months or more.”

3. P. 5 main exposure: How was missing info on BF handled? Was the last information brought forward or were the duration truncated at the last measurement available? How many measures did the kids have on average? How many were followed until age 2, 4 and 6 mo? And were kids included if they only had a measurement at week 1? The BF pattern at week 1 may not be so informative,

taking into account that almost all women initiate BF. Did children who were excluded due to missing information on BF differ from those included? Maybe these things could be described in an appendix.

Reply: We included only children with complete breastfeeding data (information on breastfeeding at 1 week, 2, 4 and 6 months).

The Statistical analysis has been revised to make this clear: "All models were restricted to observations with complete data on breastfeeding and covariates."

In total, 2127 children were excluded due to missing breastfeeding data at one or more occasions (around 5%). This should not have a significant effect on estimates. Information on children excluded due to missing data on breastfeeding or covariates is provided in table 1.

4. P. 6: Covariates: How were covariates chosen? I wonder if all of these are relevant confounders? Are they associated with BF and the outcomes? Why did the authors group maternal age and parity instead of using these as continuous variables? Similarly for year of birth.

Reply: Covariates were chosen to adjust for potential confounding of demography, socioeconomic status and perinatal conditions. All covariates were considered to be independent risk factors for infectious diseases. None of the covariates were considered to be on "the causal pathway" between breastfeeding and infectious diseases. All covariates were associated with the main outcomes (number of hospitalisations in early and later childhood). All covariates were also associated with the exposure; this was tested using logistic regression models with "exclusive breastfeeding ≥ 6 months" as a binary outcome. We agree that continuous variables could have been used for maternal age and parity. However, including maternal age and parity as continuous variables did not improve the model fit.

5. Could there be an interaction with Sex? Since boys and girls have different risks of infections in infancy? Did the authors investigate sex-specific analyses?

Reply: The risk of hospitalisations was similar for male and females e.g. the risk of hospitalisations for infectious disease in early childhood (0-2 years) is presented below.

Male Adjusted IRR* (95% CI)

No BF: 1,78 (1,19- 2,66)

BF <6 MO and EBF <4 MO: 1,43 (1,19-1,74)

BF <6 MO with EBF 4–5 MO: 1,29 (0,90-1,84)

BF ≥ 6 MO and EBF <4 MO: 1,22 (0,98-1,52)

BF ≥ 6 MO with EBF 4–5 MO: 0,94 (0,80-1,11)

EBF ≥ 6 MO: 1,00

Female Adjusted IRR* (95% CI)

No BF: 1,85 (1,23-2,80)

BF <6 MO and EBF <4 MO: 1,38 (1,16-1,64)

BF <6 MO with EBF 4–5 MO: 0,94 (0,66-1,33)

BF ≥ 6 MO and EBF <4 MO: 1,35 (1,11-1,65)

BF ≥ 6 MO with EBF 4–5 MO: 1,13 (0,97-1,31)

EBF ≥ 6 MO: 1,00

* Adjusted for year of birth small for gestational age, large for gestational age, congenital malformation, maternal age, parity, maternal smoking, maternal education level and maternal country of birth.

6. What is the number of cases for enteric and upper/lower respiratory infections? It would be preferable to have them in the manuscript, but they could also be added to the supplementary tables.

Reply: The text has been revised as suggested:

“In early childhood, the study included 1485 admissions for respiratory infections (799 admissions for upper respiratory infections and 686 admissions for lower respiratory infections), 762 admissions for enteric infections and 1080 admissions for other infections. In later childhood, the study included 630 admissions for respiratory infections (381 admissions for upper respiratory infections and 249 admissions for lower respiratory infections), 300 admissions for enteric infections and 448 admissions for other infections.”

7. P. 8 line 52-56 + P 9 line 29-31 + p. 11 line 40 + p. 12 line 25: I slightly disagree with the authors in the interpretation that, in later childhood, the risk for overall hospitalization remains higher only for children BF < 6 mo with EBF <4 mo. For the two other groups who had reduced risks in early childhood, the lower 95% CI for IRR is 1.00 – and the effect size estimate (the value most compatible with the data) is still clearly positive and not so much different from in early childhood.

Reply: The text has been revised:

“Figure 1 shows the association between breastfeeding categories and overall hospitalisations for infectious disease in early and later childhood. The risk of hospitalisations for infections decreased with duration of exclusive breastfeeding until 4 months of age. In early childhood, the highest risk was found in children who were not breastfed (aIRR 1.89 [95% CI 1.45–2.47]) and in children breastfed < 6 months and exclusive breastfeeding < 4 months (aIRR 1.41 [95% CI 1.25–1.59]). Similarly, the risk of hospitalisations for infectious diseases in later childhood was highest in children who were not breastfed (aIRR 1.52 [95% CI 1.00–2.33], controlled for previous admissions) and in children breastfed < 6 months and exclusive breastfeeding < 4 months (aIRR 1.43 [95% CI 1.16–1.75], controlled for previous admissions).”

8. P 8: I think the authors need to mention that children EBF for 4-5 mo did not have different risks than those EBF for 6 mo. This is an important finding in terms of providing evidence for BF recommendations.

Reply: The text has been revised:

“The risk of hospitalisations for infectious diseases was comparable in children exclusively breastfed 4 to 5 months and children exclusively breastfed 6 months or more.”

9. P. 8 line 56-58: It would be preferable if the authors describe what the appendix shows (here or in the appendix). Did adjustment for BMI change the results?

Reply: Sensitivity analyses are discussed in the discussion.

Sensitivity analyses revealed no substantial changes in point estimates after adjusting for pregnancy BMI, indicating no major confounding effect. Moreover, sensitivity analyses showed similar associations between breastfeeding categories and overall hospitalisations for infectious disease in the first (1998–2003) and second (2004–2010) birth cohort. Consequently, the changed definition of exclusive breastfeeding in 2004 did not significantly affect the overall risk of hospitalisation for infectious diseases.

10. P 9 line 5-12: + p. 9 line 36 + p 11 line 43 + p. 12 line 30: I do not follow why the authors only report/mention the association between BF and respiratory tract infection in later childhood. Although the associations in early childhood are weaker, they are all significant (if EBF < 4 mo) and follow the same pattern as in later childhood.

Reply: The text has been revised:

“Figure 2 shows associations between breastfeeding categories and hospitalisations for respiratory tract and enteric infections in early and later childhood. In early childhood, the risk of both respiratory and enteric infections decreased with duration of exclusive breastfeeding until 4 months of age. The strongest association was found between no breastfeeding and enteric infections (aIRR 3.32 [95% CI

2.14–5.14]). In later childhood, the risk of respiratory infections decreased with duration of exclusive breastfeeding until 4 months of age. The highest risk was observed in children who were not breastfed (aIRR 2.53 [95% CI 1.51–4.24], controlled for previous admissions).

11. P. 10: line 3-7: I do not follow how the findings from the US study contracts with the results by the authors. This study also showed an association between BF and respiratory tract infection in early childhood.

Reply: The text has been revised to make this clear:

“In contrast to our findings, a US study reported that full breastfeeding for 6 months was, in comparison with full breastfeeding for 4 months, associated with a decreased risk of respiratory tract infection during the first two years of life.”

12. Nice figures. The authors may consider showing the y-axis on logarithmic scale. Then it will be evident than an IRR of 0.5 is of similar strength as an IRR of 2.

Reply: The figures have been revised as suggested.

Minor

- P. 6: what is “LISA”?

Reply: It an acronym for “the Longitudinal integration database for health insurance and labour market studies”.

The text has been revised:

“Maternal education level at year of birth (secondary school or less [≤ 9 years], upper secondary school [10–12 years], short postsecondary education [13–14 years] and long postsecondary education [≥ 15 years]) was obtained from the Longitudinal integration database for health insurance and labour market studies”

- P. 7: Please describe the time unit for the statistical analyses.

Reply: The text has been revised:

“Log follow-up time (days) was used as an offset.”

- P. 8 and elsewhere: I think the authors mean ‘censoring’ rather than ‘censuring’.

Reply: This has been corrected.

- P. 8 line 41+: it is not clear if the authors present crude or adjusted results.

Reply: It is stated in the statistical sections that “Results were presented as adjusted incidence rate ratios (aIRRs) with 95% confidence intervals”. All IRR are adjusted IRR (aIRR) and presented as aIRR.

- P. 9 line 34 + p. 11 line 40: Do the authors mean ‘exclusive bf for at least 4 months was associated with a reduced risk of hosp. for respiratory infections..’? (i.e. add exclusive)

Reply: This has been corrected.

- p. 11 line 25: Do the authors mean “lack of any bf and enteric infections in later childhood”? there as an association between BF and respiratory tract infection in later childhood.

Reply: We mean that associations in later childhood need to be interpreted with caution due to the small number of events. The text has been revised:

"Therefore, the associations between breastfeeding and infections in later childhood need to be interpreted with caution."

- p. 37, Table G: first line: Please check the estimate of IRR=0.01 (0.01-0.01). This seems incorrect. How many cases are in the group?

There are no cases in this group, this emphasises that "the associations between breastfeeding and infections in later childhood need to be interpreted with caution."

Reviewer: 2

Comments to the Author

The authors show that hospital admissions for infection in young people are inversely associated with their breastfeeding exposure. I have some comments on the study analysis, presentation and interpretation.

1. Breastfeeding exposure is defined in terms of months of breastfeeding and months of exclusive breastfeeding, in quite a complicated way as described at the top of page 6. It needs to be stated there that the six groups ranked in the order they are defined represent increasing exposure to breastmilk, and that this can be exploited in the analysis.

Reply comments 1, 2 and 9:

There are two main reasons why breastfeeding was analysed as a categorical variable ("breastfeeding practices") instead of a continuous variable ("level of exposure to breastmilk"). First, breastfeeding has two dimensions, intensity and duration. These two dimensions cannot easily be described as a continuous variable i.e. it is unclear if "BF <6 MO with EBF 4–5 MO" is more or less "exposure to breastmilk" than "BF ≥6 MO and EBF <4 MO". Consequently, they cannot be ranked. Secondly, international recommendations combine intensity and duration e.g. "exclusive breastfeeding for at least 4 months with exclusive or predominant breastfeeding for approximately 6 months" or "breastfeeding for at least 12 months with exclusive breastfeeding for about 6 months". Our aim was to make present results that are relevant for breastfeeding recommendations. Therefore, it was necessary to analyse breastfeeding as a categorical variable.

Some previous studies have used a continuous variable ("duration of exclusive breastfeeding" or "duration of any breastfeeding"), others have used a categorical variables. A continuous variable is preferable if the aim is to establish an association between breastfeeding and infections. A categorical variable is needed if the aim is to analyse what combination of exclusive and partial breastfeeding should be recommended. The association between breastfeeding and infections is now well established. Consequently, most recent studies have used a categorical variable to analyse breastfeeding practices.

However, we agree that "picking out particular groups that are or are not significant" is not preferable. We have revised the manuscript. The focus is now on the main finding, that the risk of infections decreased with duration of exclusive breastfeeding until 4 months.

2. If the groups are numbered from 0 to 5 (or equivalently 1 to 6) then the results in Figures 1 and 2 can be tested for their association with breastmilk exposure by adding the interaction term to the model, which is evident in most of the plots. It is important to view the results "in the round" rather than picking out particular groups that are or are not significant - the trend is what matters. These results would be best presented in a table, perhaps comparing the fit of the models with a linear trend to those with the data grouped.

Reply: Please see replay comment 1.

3. Some smaller points. The Abstract Results involve lots of aIRRs which are hard to read, and presenting them to one not two decimal places would help the reader without losing important information. Perhaps also make the point there that they represent a dose-response relationship.

Reply: IRRs are normally presented with two decimal in BMJ Open. However, if the editor prefers, we will present IRR with one decimal places.

The abstract has been revised to make it clear that there is a dose-response relationship:
“The risk of hospitalisations for infections decreased with duration of exclusive breastfeeding until 4 months of age. “

4. The Introduction includes the unreferenced statement that "Breastfeeding reduces the risk of infectious diseases", which looks odd as the study aims to test the statement. I recommend omitting the sentence, and it then reads well.

Reply: This has been corrected.

5. The acronym LISA is not defined.

Reply: The text has been revised:
“the Longitudinal integration database for health insurance and labour market studies”.

6. The country groupings can be simplified to: Sweden, Other Nordic countries, Other Europe or North America, Asia, Africa and other (see page 6 and Table 1).

Reply: The text has been revised as suggested:
“Maternal country of birth (Sweden, Other Nordic, Other Europe and North America, Asia, Africa and Other) was retrieved from the Multi-Generation Register.”

7. The negative binomial models have one outcome measure and so are univariate; in addition they have multiple covariates and so are multivariable.

Reply: The text has been revised: “Crude and adjusted associations between breastfeeding practices and number of hospitalisations were estimated using negative binomial regression models”

8. In the Results several numbers have commas rather than periods as delimiter.
This has been corrected.

9. The word 'censuring' should be 'censoring' in several places. Also 'emigration' is clearer than 'international migrating'.

Reply: This has been corrected.

10. Table 1 would be easier to read with the following changes: add a Total row at the top, giving numbers and percentages in each of the six groups, and the percentages can then be compared with those in later rows to see if the group is over- or under-represented. Also the numbers and percentages would be better formatted as % (n) with the percentages given to two significant digits, i.e. whole numbers for 10% or more.

Reply: The table has been revised as suggested.

11. In Figures 1 and 2 IRR should be on a log scale, i.e. 2 and 0.5 should be equidistant from 1. This will have the effect of making the plots closer to linear and hence emphasising the linear trend.

Reply: The figures have been revised as suggested.

12. Personally I am not convinced that the multiple supplementary tables add much of value. However if they are retained they should include information about the linear trend, or even replace the grouped results with the trend.

Reply: Please see replay comment 1.

Reviewer: 3

Comments to the Author

Summary:

This observational cross-sectional study to examine the associations between breastfeeding practices and hospitalisations for infectious disease in the first five years of life. This large cohort from Sweden between 1998 and 2010 included over 37,000 full-term singleton infants. Authors concluded that exclusive breastfeeding for at least 4 months was associated with a lower risk of overall hospitalisations for infectious diseases in early childhood and Breastfeeding was associated with decreased risk of hospitalisation for respiratory illness in later childhood.

Major comments:

1. The study cohort is large from Sweden. The findings are not unique and well described in other association studies; however, this study does serve to strengthen current knowledge.
2. Strongest determinants of both breastfeeding practice and infection risk is low socioeconomic status and poverty. Covariates of study including maternal education, maternal country of birth and maternal smoking may not adequately adjust for low socioeconomic status and/or low neighbourhood equity. Recommend adding to limitations section.

Reply: The text has been revised: "Finally, due to the observational nature of this study, we cannot rule out the risk of unmeasured or residual confounding."

3. As noted in Setting, breastfeeding guidelines in Sweden changed during the study period (1998-2002 EBF 4-6 months and 2002-2010 EBF >6months). These guidelines may have affected the duration of EBF of participants, findings and interpretation of results. Recommend adjusting for periods and/or outlining in limitation section.

Reply: Sensitivity analyses included analyses stratified by year of birth (1998 to 2003 and 2004 to 2010).

Minor comments:

1. Study may not be generalisable to other populations; especially since Sweden has relatively high breastfeeding rates, free primary and health care, and generous parental leave scheme, compared to other industrialized countries. Recommend outlining in limitation section.

Reply: The text has been revised: "Moreover, Sweden has family-friendly policies that enable breastfeeding including a generous parental leave scheme. Consequently, our findings may not be generalised to other populations with less family-friendly policies and lower breastfeeding rates."

2. Other limitations may include unmeasured confounders, such as seasonal and cyclical variation of infectious disease (Inparticular RSV which impacts young children) and Indigenous peoples of Sweden (may have lower BF rates and increased rates of RSV infection).

Reply: The text has been revised: “Finally, due to the observational nature of this study, we cannot rule out the risk of unmeasured or residual confounding.”

VERSION 2 – REVIEW

REVIEWER	Bjerregaard, Lise Center for Clinical Research and Disease Prevention, Bispebjerg and Frederiksberg Hospital
REVIEW RETURNED	16-Mar-2021

GENERAL COMMENTS	The authors have done a great job in answering the comments of the reviewers and revising the manuscript. I still have a few minor comments left. Abstract  1. It is not clear who the reference group is. Moreover, from the comparison of 4-5 mo vs 6 it seems as it is the group EBF 4-5 mo, but actually it is EBF 6 mo. I suggest the authors revise here and in the discussion. 2. The part on strengths and limitations are mostly a summary of the study. Strength or limitations could be highlighted more. Results:  3. P. 10: Who is the reference group? 4. What did the sensitivity analysis show? If anything? And what is the rationale for adjusting for maternal BMI separately? I think this belongs in the results section. 5. The authors provided sex specific results only in the rebuttal. They may add a brief comment about this in the manuscript. Discussion  6. Who are the reference groups in the studies that are cited/compared to? Tables  7. Table 1: SGA/LGA: the labels for ‘yes’ and ‘no’ are swapped (yes are on the lower row). 8. Table G: The authors should consider not showing a result of 0.01 (0.01, 0.01). How many children does it represent? Is the estimate valid?
--

REVIEWER	Cole, Tim UCL Institute of Child Health
REVIEW RETURNED	19-Mar-2021

GENERAL COMMENTS	The authors have responded positively to my previous comments. However there are a few additional points.  1. The study's research question only becomes clear in the second paragraph of the Discussion, where it says "However, the optimal breastfeeding practices is under discussion." This clarifies that the purpose of the study is to see whether there is a difference in infection rate associated with exclusive breastfeeding to 4 months as against 6 months. The title, abstract and introduction do not explicitly state this as the aim, and making it clear from the outset would strengthen the paper. I recommend changing the title to something like "Breastfeeding
--

	practice, breastfeeding policy and childhood infections: a cohort study" [note that "practice" is preferable to "practices"], plus extending the abstract aim to inform policy, and changing the abstract conclusion to bring out the policy implications of the findings. 2. Breastfeeding practice is grouped into six groups, which are ranked in the figures by overall breastfeeding and then exclusive breastfeeding. This leads to a pattern whereby the fourth group has consistently higher IRRs than its neighbours, suggesting it is ranked incorrectly. Now if the ranking were done instead by exclusive breastfeeding and then overall breastfeeding, this would rank the fourth group as third, and it would strengthen the dose-response relation in five of the six figures. It would also fit with the study aim, to show that it is exclusive breastfeeding not overall breastfeeding that impacts most on infection. 3. It would be useful to define what is meant by a "full term singleton". For example does a 43-week-gestation infant count? 4. Some tiny wording points. Page 7 line 21, it should be criterion (singular) not criteria (plural). Page 12 top line, "decreased risk of ...", and "Sensitivity analyses" twice in the next paragraph. 5. In the figures, relying on colour to distinguish between groups means that print readers with black-and-white will miss out. I suggest using symbols as well as colour to distinguish between groups. Tim Cole
--	---

VERSION 2 – AUTHOR RESPONSE

Reviewer: 1

Dr. Lise Bjerregaard, Center for Clinical Research and Disease Prevention, Bispebjerg and Frederiksberg Hospital

Comments to the Author:

The authors have done a great job in answering the comments of the reviewers and revising the manuscript. I still have a few minor comments left.

Abstract

1. It is not clear who the reference group is. Moreover, from the comparison of 4-5 mo vs 6 it seems as it is the group EBF 4-5 mo, but actually it is EBF 6 mo. I suggest the authors revise here and in the discussion.

Reply: The text has been revised as suggested:

Abstract: "The risk of hospitalisations for infectious diseases decreased with duration of exclusive breastfeeding until 4 months of age. In early childhood, breastfeeding was associated with a decreased risk of enteric and respiratory infections. In comparison with exclusive breastfeeding 6 months or more, the strongest association was found between no breastfeeding and enteric infections (aIRR 3.32 [95% CI 2.14–5.14]). In later childhood, breastfeeding was associated with a lower risk of

respiratory infections. In comparison with children exclusively breastfed 6 months or more, the highest risk was found in children who were not breastfed (aIRR 2.53 [95% CI 1.51–4.24]).”

Discussion: “In comparison with exclusive breastfeeding 6 months or more, exclusive breastfeeding 4 to 5 months was not associated with a higher risk of enteric or respiratory infections.”

2. The part on strengths and limitations are mostly a summary of the study. Strength or limitations could be highlighted more.

Reply: The text has been revised as suggested:

“This register-based cohort study included almost all (<10% missing data) term and post-term singletons born in a geographical region over 13 years, which reduces the risk of selection bias. Children were followed from birth until 5 years of age or censoring, which enabled us to examine the effect of breastfeeding in both early (<2 years) and later childhood (2–4 years). Information on pregnancy, birth and sociodemographic characteristics were obtained from high-quality health and administrative registers, which allowed us to adjust analyses for several potential confounders including congenital malformation, large/small for gestational age, maternal age, maternal smoking, parity, maternal education level and maternal country of birth. Infections were identified using de-identified hospital discharge data, cases could not be confirmed by information obtained from medical records. Breastfeeding practice was not continuously collected, information on breastfeeding practice was only available at 1 week, 2, 4 and 6 months.”

Results:

3. P. 10: Who is the reference group?

Reply: The text has been revised as suggested:

“In comparison with children exclusively breastfed 6 months or more, the highest risk of hospitalisations for infectious diseases in early childhood was found in children who were not breastfed (aIRR 1.89 [95% CI 1.45–2.47]) and in children exclusive breastfed <4 months with breastfeeding <6 months (aIRR 1.41 [95% CI 1.25–1.59]).” “In comparison with exclusive breastfeeding 6 months or more, the strongest association was found between no breastfeeding and enteric infections (aIRR 3.32 [95% CI 2.14–5.14]). In later childhood, the risk of respiratory infections decreased with duration of exclusive breastfeeding until 4 months of age. In comparison with children exclusively breastfed 6 months or more, the highest risk of respiratory infections was observed in children who were not breastfed (aIRR 2.53 [95% CI 1.51–4.24], controlled for previous admissions).”

RESULTS

4. What did the sensitivity analysis show? If anything? And what is the rationale for adjusting for maternal BMI separately? I think this belongs in the results section.

Reply: The text has been revised, sensitivity analyses are now presented in the results section. The rationale for adjusting for maternal BMI separately is explained in statistical analysis “Maternal BMI was excluded from the original analyses due to a large proportion of missing data (12%).”

5. The authors provided sex specific results only in the rebuttal. They may add a brief comment about this in the manuscript.

Reply: The text has been revised as suggested: “In sex-stratified analyses, associations between breastfeeding categories and overall hospitalisations for infectious disease were similar in boys and girls (data not shown).”

Discussion

6. Who are the reference groups in the studies that are cited/compared to?

Reply: The text has been revised as suggested: Reference groups are now presented.

Tables

7. Table 1: SGA/LGA: the labels for ‘yes’ and ‘no’ are swapped (yes are on the lower row).

Reply: This has been corrected.

8. Table G: The authors should consider not showing a result of 0.01 (0.01, 0.01). How many children does it represent? Is the estimate valid?

Reply: The table has been revised, the result has been removed and this is explained "The estimate for "No BF" in 2004-2010 is not presented due to few events."

Reviewer: 2

Dr. Tim Cole, UCL Institute of Child Health

Comments to the Author:

The authors have responded positively to my previous comments. However there are a few additional points.

1. The study's research question only becomes clear in the second paragraph of the Discussion, where it says "However, the optimal breastfeeding practices is under discussion." This clarifies that the purpose of the study is to see whether there is a difference in infection rate associated with exclusive breastfeeding to 4 months as against 6 months. The title, abstract and introduction do not explicitly state this as the aim, and making it clear from the outset would strengthen the paper.

I recommend changing the title to something like "Breastfeeding practice, breastfeeding policy and childhood infections: a cohort study" [note that "practice" is preferable to "practices"], plus extending the abstract aim to inform policy, and changing the abstract conclusion to bring out the policy implications of the findings.

Reply: The text has been revised as suggested:

Title "Breastfeeding practice, breastfeeding policy and hospitalisations for infectious diseases in early and later childhood: a register-based study in Uppsala County, Sweden"

Abstract "Objective: To examine the association between breastfeeding practice and hospitalisations for infectious diseases in early and later childhood, i.e. to compare exclusive breastfeeding 4 to 5 months with exclusive breastfeeding 6 months or more. Thereby, provide evidence to inform breastfeeding policy."... "Conclusions: Our results support breastfeeding guidelines that recommend exclusive breastfeeding for at least 4 months."

Introduction "The aim of this study was to examine the association between breastfeeding practice and hospitalisations for infectious diseases in early (<2 years) and later childhood (2–4 years), i.e. to compare exclusive breastfeeding 4 to 5 month with exclusive breastfeeding 6 months or more"

2. Breastfeeding practice is grouped into six groups, which are ranked in the figures by overall breastfeeding and then exclusive breastfeeding. This leads to a pattern whereby the fourth group has consistently higher IRRs than its neighbours, suggesting it is ranked incorrectly.

Now if the ranking were done instead by exclusive breastfeeding and then overall breastfeeding, this would rank the fourth group as third, and it would strengthen the dose-response relation in five of the six figures. It would also fit with the study aim, to show that it is exclusive breastfeeding not overall breastfeeding that impacts most on infection.

Reply: Breastfeeding categories are now ranked as suggested, figures and tables have been updated. The method section has been revised.

"Thereafter, exclusive breastfeeding was considered to be present until "partial breastfeeding" or "no breastfeeding" was reported; breastfeeding was considered to be present until "no breastfeeding" was reported e.g. breastfeeding was categorised as "exclusive breastfeeding 4–5 months with breastfeeding \geq 6 months" if "exclusive breastfeeding" was reported at 1 week, 2 months, 4 months and "partial breastfeeding" was reported at 6 months. Information on breastfeeding intensity and duration was combined into the following categories: No breastfeeding (No BF), exclusive

breastfeeding <4 months with breastfeeding <6 months (EBF <4 MO with BF <6 MO), exclusive breastfeeding <4 months with breastfeeding ≥6 months (EBF <4 MO with BF ≥6 MO), exclusive breastfeeding 4–5 months with breastfeeding <6 months (EBF 4–5 MO with BF <6 MO), exclusive breastfeeding 4–5 months with breastfeeding ≥6 months (EBF 4–5 MO with BF ≥6 MO) and exclusive breastfeeding ≥6 months (EBF ≥6 MO).“

3. It would be useful to define what is meant by a "full term singleton". For example does a 43-week-gestation infant count?

Reply: The text has been revised: "term and post-term (gestational age 37 weeks or more)"

4. Some tiny wording points. Page 7 line 21, it should be criterion (singular) not criteria (plural). Page 12 top line, "decreased risk of ...", and "Sensitivity analyses" twice in the next paragraph.

Reply: This has been corrected.

5. In the figures, relying on colour to distinguish between groups means that print readers with black-and-white will miss out. I suggest using symbols as well as colour to distinguish between groups.

Reply: Figures have been revised as suggested: "Crude analyses (blue circles), adjusted analyses (red squares) and adjusted analyses controlled for previous admissions (green triangles)".

Tim Cole

Reviewer: 1

Competing interests of Reviewer: None declared

Reviewer: 2

Competing interests of Reviewer: None declared

VERSION 3 – REVIEW

REVIEWER	Cole, Tim UCL Institute of Child Health
REVIEW RETURNED	06-Apr-2021
GENERAL COMMENTS	N/A